# Certified Adversarial Robustness
# with Additive Noise

**Bai Li**
Department of Statistical Science
Duke University
bai.li@duke.edu

**Changyou Chen**
Department of CSE
University at Buffalo, SUNY
cchangyou@gmail.com

**Wenlin Wang**
Department of ECE
Duke University
wenlin.wang@duke.edu

**Lawrence Carin**
Department of ECE
Duke University
lcarin@duke.edu

## Abstract

The existence of adversarial data examples has drawn significant attention in the deep-learning community; such data are seemingly minimally perturbed relative to the original data, but lead to very different outputs from a deep-learning algorithm. Although a significant body of work on developing defensive models has been considered, most such models are heuristic and are often vulnerable to adaptive attacks. Defensive methods that provide theoretical robustness guarantees have been studied intensively, yet most fail to obtain non-trivial robustness when a large-scale model and data are present. To address these limitations, we introduce a framework that is scalable and provides certified bounds on the norm of the input manipulation for constructing adversarial examples. We establish a connection between robustness against adversarial perturbation and additive random noise, and propose a training strategy that can significantly improve the certified bounds. Our evaluation on MNIST, CIFAR-10 and ImageNet suggests that the proposed method is scalable to complicated models and large data sets, while providing competitive robustness to state-of-the-art provable defense methods.

## 1 Introduction

Although deep neural networks have achieved significant success on a variety of challenging machine learning tasks, including state-of-the-art accuracy on large-scale image classification [1, 2], the discovery of adversarial examples [3] has drawn attention and raised concerns. Adversarial examples are carefully perturbed versions of the original data that successfully fool a classifier. In the image domain, for example, adversarial examples are images that have no visual difference from natural images, but that lead to different classification results [4].

A large body of work has been developed on defensive methods to tackle adversarial examples, yet most remain vulnerable to adaptive attacks [3–10]. A major drawback of many defensive models is that they are heuristic and fail to obtain a theoretically justified guarantee of robustness. On the other hand, many works have focused on providing provable/certified robustness of deep neural networks [11–17].

Recently, [18] provided theoretical insight on certified robust prediction, building a connection between differential privacy and model robustness. It was shown that adding properly chosen noise to the classifier will lead to certified robust prediction. Building on ideas in [18], we conduct an analysis of model robustness based on Rényi divergence [19] between the outputs of models for

natural and adversarial examples when random noise is added, and show a higher upper bound on the tolerable size of perturbations compared to [18]. In addition, our analysis naturally leads to a connection between adversarial defense and robustness to random noise. Based on this, we introduce a comprehensive framework that incorporates stability training for additive random noise, to improve classification accuracy and hence the certified bound. Our contributions are as follows:

- We derive a certified bound for robustness to adversarial attacks, applicable to models with general activation functions and network structures. Specifically, according to [20], the derived bound for $\ell_1$ perturbation is tight in the binary case.

- Using our derived bound, we establish a strong connection between robustness to adversarial perturbations and additive random noise. We propose a new training strategy that accounts for this connection. The new training strategy leads to significant improvement on the certified bounds.

- We conduct a comprehensive set of experiments to evaluate both the theoretical and empirical performance of our methods, with results that are competitive with the state of the art.

## 2 Background and Related Work

Much research has focused on providing provable/certified robustness for neural networks. One line of such studies considers distributionally robust optimization, which aims to provide robustness to changes of the data-generating distribution. For example, [21] study robust optimization over a $\phi$-divergence ball of a nominal distribution. A robustness with respect to the Wasserstein distance between natural and adversarial distributions was provided in [22]. One limitation of distributional robustness is that the divergence between distributions is rarely used as an empirical measure of strength of adversarial attacks.

Alternatively, studies have attempted to provide a certified bound of minimum distortion. In [11] a certified bound is derived with a small loss in accuracy for robustness to $\ell_2$ perturbations in two-layer networks. A method based on semi-definite relaxation is proposed in [12] for calculating a certified bound, yet their analysis cannot be applied to networks with more than one hidden layer. A robust optimization procedure is developed in [13] by considering a convex outer approximation of the set of activation functions reachable through a norm-bounded perturbation. Their analysis, however, is still limited to ReLU networks and pure feedforward networks. Algorithms to efficiently compute a certified bound are considered in [14] by utilizing the property of ReLU networks as well. Recently, their idea has been extended by [15] and relaxed to general activation functions. However, both of their analyses only apply to the multi-layer perceptron (MLP), limiting the application of their results.

In general, most analyses for certified bounds rely on the properties of specific activation functions or model structures, and are difficult to scale. Several works aim to generalize their analysis to accommodate flexible model structures and large-scale data. For example, [16] formed an optimization problem to obtain an upper bound via Lagrangian relaxation. They successfully obtained the first non-trivial bound for the CIFAR-10 data set. The analysis of [13] was improved in [17], where it was scaled up to large neural networks with general activation functions, obtaining state-of-the-art results on MNIST and CIFAR-10. Certified robustness is obtained by [18] by analyzing the connection between adversarial robustness and differential privacy. Similar to [16, 17], their certified bound is agnostic to model structure and is scalable, but it is loose and is not comparable to [16, 17]. Our approach maintains all the advantages of [18], and significantly improves the certified bound with more advanced analysis.

The connection between adversarial robustness and robustness to added random noise has been studied in several works. In [23] this connection is established by exploring the curvature of the classifier decision boundary. Later, [24] showed adversarial robustness requires reducing error rates to essentially zero under large additive noise. While most previous works use concentration of measure as their analysis tool, we approach such connection from a different perspective using Rényi divergence [19]; we illustrate the connection to robustness to random noise in a more direct manner. More importantly, our analysis suggests improving robustness to additive Gaussian noise can directly result in the improvement of the certified bound.

# 3 Preliminaries

## 3.1 Notation

We consider the task of image classification. Natural images are represented by $\mathbf{x} \in \mathcal{X} \triangleq [0, 1]^{h \times w \times c}$, where $\mathcal{X}$ represents the image space, with $h, w$ and $c$ denoting the height, width, and number of channels of an image, respectively. An image classifier over $k$ classes is considered as a function $f : \mathcal{X} \to \{1, \ldots, k\}$. We only consider classifiers constructed by deep neural networks (DNNs). To present our framework, we define a stochastic classifier, a function $f$ over $\mathbf{x}$ with output $f(\mathbf{x})$ being a multinomial distribution over $\{1, \ldots, k\}$, i.e., $P(f(\mathbf{x}) = i) = p_i$ for $\sum_i p_i = 1$. One can classify $\mathbf{x}$ by picking $\operatorname{argmax}_i p_i$. Note this distribution is different from the one generated from softmax.

## 3.2 Rényi Divergence

Our theoretical result depends on the Rényi divergence, defined in the following [19].

**Definition 1 (Rényi Divergence)** *For two probability distributions $P$ and $Q$ over $\mathcal{R}$, the Rényi divergence of order $\alpha > 1$ is*

$$D_\alpha(P\|Q) = \frac{1}{\alpha - 1} \log \mathbb{E}_{x \sim Q} \left( \frac{P}{Q} \right)^\alpha \tag{1}$$

## 3.3 Adversarial Examples

Given a classifier $f : \mathcal{X} \to \{1, \ldots, k\}$ for an image $\mathbf{x} \in \mathcal{X}$, an adversarial example $\mathbf{x}'$ satisfies $\mathcal{D}(\mathbf{x}, \mathbf{x}') < \epsilon$ for some small $\epsilon > 0$, and $f(\mathbf{x}) \neq f(\mathbf{x}')$, where $\mathcal{D}(\cdot, \cdot)$ is some distance metric, i.e., $\mathbf{x}'$ is close to $\mathbf{x}$ but yields a different classification result. The distance is often described in terms of an $\ell_p$ metric, and in most of the literature $\ell_2$ and $\ell_\infty$ metrics are considered. In our development, we focus on the $\ell_2$ metric but also provide experimental results for $\ell_\infty$. More general definitions of adversarial examples are considered in some works [25], but we only address norm-bounded adversarial examples in this paper.

## 3.4 Adversarial Defense

Classification models that are robust to adversarial examples are referred to as adversarial defense models. We introduce the most advanced defense models in two categories.

Empirically, the most successful defense model is based on adversarial training [4, 26], that is augmenting adversarial examples during training to help improve model robustness. TRadeoff-inspired Adversarial DEfense via Surrogate-loss minimization (TRADES) [27] is a variety of adversarial training that introduces a regularization term for adversarial examples:

$$\mathcal{L} = \mathcal{L}(f(\mathbf{x}), y) + \gamma \mathcal{L}(f(\mathbf{x}, f(\mathbf{x}_{\text{adv}}))$$

where $\mathcal{L}(\cdot, \cdot)$ is the cross-entropy loss. This defense model won 1st place in the NeurIPS 2018 Adversarial Vision Challenge (Robust Model Track) and has shown better performance compared to previous models [26].

On the other hand, the state-of-the-art approach for provable robustness is proposed by [17], where a dual network is considered for computing a bound for adversarial perturbation using linear-programming (LP), as in [13]. They optimize the bound during training to achieve strong provable robustness.

Although empirical robustness and provable robustness are often considered as orthogonal research directions, we propose an approach that provides both. In our experiments, presented in Sec. 6, we show our approach is competitive with both of the aforementioned methods.

# 4 Certified Robust Classifier

We propose a framework that yields an upper bound on the tolerable size of attacks, enabling certified robustness on a classifier. Intuitively, our approach adds random noise to pixels of adversarial examples before classification, to eliminate the effects of adversarial perturbations.

---
**Algorithm 1** Certified Robust Classifier

---
**Require:** An input image $\mathbf{x}$; A standard deviation $\sigma > 0$; A classifier $f$ over $\{1, \ldots, k\}$; Number of iterations $n$ ($n = 1$ is sufficient if only the robust classification $c$ is desired).
1: Set $i = 1$.
2: **for** $i \in [n]$ **do**
3:     Add i.i.d. Gaussian noise $N(0, \sigma^2)$ to each pixel of $\mathbf{x}$ and apply the classifier $f$ on it. Let the output be $c_i = f(\mathbf{x} + N(\mathbf{0}, \sigma^2 I))$.
4: **end for**
5: Estimate the distribution of the output as $p_j = \frac{\#\{c_i = j : i = 1, \ldots, n\}}{n}$.
6: Calculate the upper bound:

$$L = \sup_{\alpha > 1} \left( -\frac{2\sigma^2}{\alpha} \log \left( 1 - p_{(1)} - p_{(2)} + 2 \left( \frac{1}{2} \left( p_{(1)}^{1-\alpha} + p_{(2)}^{1-\alpha} \right) \right)^{\frac{1}{1-\alpha}} \right) \right)^{1/2}$$

where $p_{(1)}$ and $p_{(2)}$ are the first and the second largest values in $p_1, \ldots, p_k$.
7: Return classification result $c = \text{argmax}_i\, p_i$ and the tolerable size of the attack $L$.

---

Our approach is summarized in Algorithm 1. In the following, we develop theory to prove the certified robustness of the proposed algorithm. Our goal is to show that if the classification of $\mathbf{x}$ in Algorithm 1 is in class $c$, then for any examples $\mathbf{x}'$ such that $\|\mathbf{x} - \mathbf{x}'\|_2 \leq L$, the classification of $\mathbf{x}'$ is also in class $c$.

To prove our claim, first recall that a stochastic classifier $f$ over $\{1, \ldots, k\}$ has an output $f(\mathbf{x})$ corresponding to a multinomial distribution over $\{1, \ldots, k\}$, with probabilities as $(p_1, \ldots, p_k)$. In this context, robustness to an adversarial example $\mathbf{x}'$ generated from $\mathbf{x}$ means $\text{argmax}_i\, p_i = \text{argmax}_j\, p'_j$ with $P(f(\mathbf{x}) = i) = p_i$ and $P(f(\mathbf{x}') = j) = p'_j$, where $P(\cdot)$ denotes the probability of an event. In the remainder of this section, we show Algorithm 1 achieves such robustness based on the Rényi divergence, starting with the following lemma.

**Lemma 1** *Let $P = (p_1, \ldots, p_k)$ and $Q = (q_1, \ldots, q_k)$ be two multinomial distributions over the same index set $\{1, \ldots, k\}$. If the indices of the largest probabilities do not match on $P$ and $Q$, that is $\text{argmax}_i\, p_i \neq \text{argmax}_j\, q_j$, then*

$$D_\alpha(Q\|P) \geq -\log \left( 1 - p_{(1)} - p_{(2)} + 2 \left( \frac{1}{2} \left( p_{(1)}^{1-\alpha} + p_{(2)}^{1-\alpha} \right) \right)^{\frac{1}{1-\alpha}} \right)$$

*where $p_{(1)}$ and $p_{(2)}$ are the largest and the second largest probabilities among the set of all $p_i$.*

To simplify notation, we define $M_p(x_1, \ldots, x_n) = \left( \frac{1}{n} \sum_{i=1}^{n} x_i^p \right)^{1/p}$ as the generalized mean. The right hand side (RHS) of the condition in Lemma 1 then becomes $-\log \left( 1 - 2M_1\left( p_{(1)}, p_{(2)} \right) + 2M_{1-\alpha}\left( p_{(1)}, p_{(2)} \right) \right)$.

Lemma 1 proposes a lower bound of the Rényi divergence for changing the index of the maximum of $P$, *i.e.*, for any distribution $Q$, if $D_\alpha(Q\|P) < -\log \left( 1 - 2M_1\left( p_{(1)}, p_{(2)} \right) + 2M_{1-\alpha}\left( p_{(1)}, p_{(2)} \right) \right)$, the index of the maximum of $P$ and $Q$ must be the same. Based on Lemma 1, we obtain our main theorem on certified robustness as follows, validating our claim.

**Theorem 2** *Suppose we have $\mathbf{x} \in \mathcal{X}$, and a potential adversarial example $\mathbf{x}' \in \mathcal{X}$ such that $\|\mathbf{x} - \mathbf{x}'\|_2 \leq L$. Given a $k$-classifier $f : \mathcal{X} \to \{1, \ldots, k\}$, let $f(\mathbf{x} + N(\mathbf{0}, \sigma^2 I)) \sim (p_1, \ldots, p_k)$ and $f(\mathbf{x}' + N(\mathbf{0}, \sigma^2 I)) \sim (p'_1, \ldots, p'_k)$.*

*If the following condition is satisfied, with $p_{(1)}$ and $p_{(2)}$ being the first and second largest probabilities in $\{p_i\}$:*

$$\sup_{\alpha > 1} -\frac{2\sigma^2}{\alpha} \log \left( 1 - 2M_1\left( p_{(1)}, p_{(2)} \right) + 2M_{1-\alpha}\left( p_{(1)}, p_{(2)} \right) \right) \geq L^2$$

*then $\text{argmax}_i\, p_i = \text{argmax}_j\, p'_j$*

The conclusion of Theorem 2 can be extended to the $\ell_1$ case by replacing Gaussian with Laplacian noise. Specifically, notice the Renyi divergence between two Laplacian distribution $\Lambda(x, \lambda)$ and $\Lambda(x', \lambda)$ is

$$\frac{1}{\alpha - 1} \log \left( \frac{\alpha}{2\alpha - 1} \exp \left( \frac{(\alpha - 1)\|x - x'\|_1}{\lambda} \right) + \frac{\alpha - 1}{2\alpha - 1} \exp \left( -\frac{\alpha\|x - x'\|_1}{\lambda} \right) \right) \xrightarrow{\alpha \to \infty} \frac{\|x - x'\|_1}{\lambda}$$

Meanwhile, $-\log \left( 1 - 2M_1 \left( p_{(1)}, p_{(2)} \right) + 2M_{1-\alpha} \left( p_{(1)}, p_{(2)} \right) \right) \xrightarrow{\alpha \to \infty} -\log(1 - p_{(1)} + p_{(2)})$, thus we have the upper bound for the $\ell_1$ perturbation:

**Theorem 3** *In the same setting as in Theorem 2, with $\|\mathbf{x} - \mathbf{x}'\|_1 \leq L$, let $f(\mathbf{x} + \Lambda(\mathbf{0}, \lambda)) \sim (p_1, \ldots, p_k)$ and $f(\mathbf{x}' + \Lambda(\mathbf{0}, \lambda)) \sim (p'_1, \ldots, p'_k)$. If $-\lambda \log(1 - p_{(1)} + p_{(2)}) \geq L$ is satisfied, then $argmax_i\, p_i = argmax_j\, p'_j$.*

In the rest of this paper, we focus on the $\ell_2$ norm with Gaussian noise, but most conclusions are also applicable to $\ell_1$ norm with Laplacian noise. A more comprehensive analysis for $\ell_1$ norm can be found in [20]. Interestingly, they have proved that the bound $-\lambda \log(1 - p_{(1)} + p_{(2)})$ is tight in the binary case for $\ell_1$ norm [20].

With Theorem 2, we can enable certified $\ell_2$ ($\ell_1$) robustness on any classifier $f$ by adding i.i.d. Gaussian (Laplacian) noise to pixels of inputs during testing, as done in Algorithm 1. It provides an upper bound for the tolerable size of attacks for a classifier, *i.e.*, as long as the pertubation size is less than the upper bound (the "sup" part in Theorem 2), any adversarial sample can be classified correctly.

**Confidence Interval and Sample Size**  In practice we can only estimate $p_{(1)}$ and $p_{(2)}$ from samples, thus the obtained lower bound is not precise and requires adjustment. Note that $(p_1, \ldots, p_k)$ forms a multinomial distribution, and therefore the confidence intervals for $p_{(1)}$ and $p_{(2)}$ can be estimated using one-sided Clopper-Pearson interval along with Bonferroni correction. We refer to [18] for further details. In all our subsequent experiments, we use the end points (lower for $p_{(1)}$ and upper for $p_{(2)}$) of the $95\%$ confidence intervals for estimating $p_{(1)}$ and $p_{(2)}$, and multiply $95\%$ for the corresponding accuracy. Moreover, the estimates for the confidence intervals are more precise when we increase the sample size $n$, but at the cost of extra computational burden. In practice, we find a sample size of $n = 100$ is sufficient.

**Choice of $\sigma$**  The formula of our lower bound indicates a higher standard deviation $\sigma$ results in a higher bound. In practice, however, a larger amount of added noise also leads to higher classification error and a larger gap between $p_{(1)}$ and $p_{(2)}$, which gives a lower bound. Therefore, the best choice of $\sigma^2$ is not obvious. We will demonstrate the effect of different choices of $\sigma^2$ in the experiments of Sec. 6.

## 5   Improved Certified Robustness

Based on the property of *generalized mean*, one can show that the upper bound is larger when the difference between $p_{(1)}$ and $p_{(2)}$ becomes larger. This is consistent with the intuition that a larger difference between $p_{(1)}$ and $p_{(2)}$ indicates more confident classification. In other words, more confident and accurate prediction in the presence of additive Gaussian noise, in the sense that $p_{(1)}$ is much larger than $p_{(2)}$, leads to better certified robustness. To this end, a connection between robustness to adversarial examples and robustness to added random noise has been established by our analysis.

Such a connection is beneficial, because robustness to additive Gaussian noise is much easier to achieve than robustness to carefully crafted adversarial examples. Consequently, it is natural to consider improving the adversarial robustness of a model by first improving its robustness to added random noise. In the context of Algorithm 1, we aim to improve the robustness of $f$ to additive random noise. Note improving robustness to added Gaussian noise as a way of improving adversarial robustness has been proposed by [28] and was later shown ineffective [29]. Our method is different

in that it requires added Gaussian noise during the testing phase, and more importantly it is supported theoretically.

There have been notable efforts at developing neural networks that are robust to added random noise [30, 31]; yet, these methods failed to defend against adversarial attacks, as they are not particularly designed for this task. Within our framework, since Algorithm 1 has no constraint on the classifier $f$, which gives the flexibility to modify $f$, we can adapt these methods to improve the accuracy of classification when Gaussian noise is present, hence improving the robustness to adversarial attacks. In this paper, we only discuss stability training, but a much wider scope of literature exists for robustness to added random noise [30, 31].

## 5.1 Stability Training

The idea of introducing perturbations during training to improve model robustness has been studied widely. In [32] the authors considered perturbing models as a construction of pseudo-ensembles, to improve semi-supervised learning. More recently, [33] used a similar training strategy, named stability training, to improve classification robustness on noisy images.

For any natural image $\mathbf{x}$, stability training encourages its perturbed version $\mathbf{x}'$ to yield a similar classification result under a classifier $f$, $i.e.$, $D(f(\mathbf{x}), f(\mathbf{x}'))$ is small for some distance measure $D$. Specifically, given a loss function $\mathcal{L}^*$ for the original classification task, stability training introduces a regularization term $\mathcal{L}(\mathbf{x}, \mathbf{x}') = \mathcal{L}^* + \gamma \mathcal{L}_{\text{stability}}(\mathbf{x}, \mathbf{x}') = \mathcal{L}^* + \gamma D(f(\mathbf{x}), f(\mathbf{x}'))$, where $\gamma$ controls the strength of the stability term. As we are interested in a classification task, we use cross-entropy as the distance $D$ between $f(\mathbf{x})$ and $f(\mathbf{x}')$, yielding the stability loss $\mathcal{L}_{\text{stability}} = -\sum_j P(y_j|\mathbf{x}) \log P(y_j|\mathbf{x}')$, where $P(y_j|\mathbf{x})$ and $P(y_j|\mathbf{x}')$ are the probabilities generated after softmax. In this paper, we add i.i.d. Gaussian noise to each pixel of $\mathbf{x}$ to construct $\mathbf{x}'$, as suggested in [33].

Stability training is in the same spirit as adversarial training, but is only designed to improve the classification accuracy under a Gaussian perturbation. Within our framework, we can apply stability training to $f$, to improve the robustness of Algorithm 1 against adversarial perturbations. We call the resulting defense method Stability Training with Noise (STN).

**Adversarial Logit Pairing**  Adversarial Logit Pairing (ALP) was proposed in [34]; it adds $D(f(\mathbf{x}), f(\mathbf{x}'))$ as the regularizer, with $\mathbf{x}'$ being an adversarial example. Subsequent work has shown ALP fails to obtain adversarial robustness [35]. Our method is different from ALP and any other regularizer-based approach, as the key component in our framework is the added Gaussian noise at the testing phase of Algorithm 1, while stability training is merely a technique for improving the robustness further. We do not claim stability training alone yields adversarial robustness.

## 6 Experiments

We perform experiments on the MNIST and CIFAR-10 data sets, to evaluate the theoretical and empirical performance of our methods. We subsequently also consider the larger ImageNet dataset. For the MNIST data set, the model architecture follows the models used in [36], which contains two convolutional layers, each containing 64 filters, followed with a fully connected layer of size 128. For the CIFAR-10 dataset, we use a convolutional neural network with seven convolutional layers along with MaxPooling. In both datasets, image intensities are scaled to $[0, 1]$, and the size of attacks are also rescaled accordingly. For reference, a distortion of $0.031$ in the $[0, 1]$ scale corresponds to $8$ in $[0, 255]$ scale. The source code can be found at `https://github.com/Bai-Li/STN-Code`.

### 6.1 Theoretical Bound

With Algorithm 1, we are able to classify a natural image $\mathbf{x}$ and calculate an upper bound for the tolerable size of attacks $L$ for this particular image. Thus, with a given size of the attack $L^*$, the classification must be robust if $L^* < L$. If a natural example is correctly classified and robust for $L^*$ simultaneously, any adversarial examples $\mathbf{x}'$ with $\|\mathbf{x} - \mathbf{x}'\|_2 < L^*$ will be classified correctly. Therefore, we can determine the proportion of such examples in the test set as **a lower bound of accuracy** given size $L^*$.

We plot in Figure 1 different lower bounds for various choices of $\sigma$ and $L^*$, for both MNIST and CIFAR-10. To interpret the results, for example on MNIST, when $\sigma = 0.7$, Algorithm 1 achieves at least $51\%$ accuracy under any attack whose $\ell_2$-norm size is smaller than $1.4$.

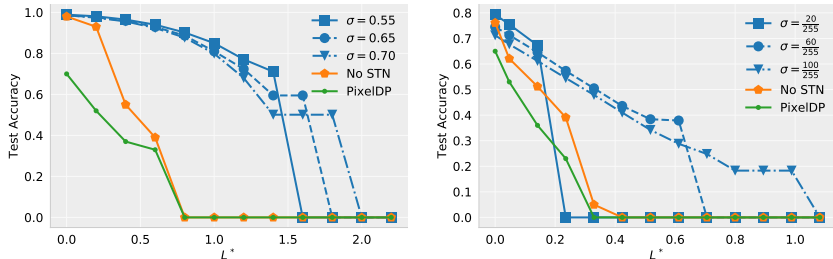

Figure 1: Accuracy lower bounds for **MNIST** (left) and **CIFAR-10** (right). We test various choices of $\sigma$ in Algorithm 1. For reference, we include results for PixelDP (green) and the lower bound without stability training (orange).

When $\sigma$ is fixed, there exists a threshold beyond which the certified lower bound degenerates to zero. The larger the deviation $\sigma$ is, the later the degeneration happens. However, larger $\sigma$ also leads to a worse bound when $L^*$ is small, as the added noise reduces the accuracy of classification.

As an ablation study, we include the corresponding lower bound without stability training. The improvement due to stability training is significant. In addition, to demonstrate the improvement of the certified bound compared to PixelDP [18], we also include the results for PixelDP. Although PixelDP also has tuning parameters $\delta$ and $\epsilon$ similar to $\sigma$ in our setting, we only include the optimal pair of parameters found by grid search, for simplicity of the plots. One observes the accuracy lower bounds for PixelDP (green) are dominated by our bounds.

We also compare STN with the approach from [17]. Besides training a single robust classifier, [17] also proposed a strategy of training a sequence of robust classifiers as a cascade model which results in better provable robustness, although it reduces the accuracy on natural examples. We compare both to STN in Table 1. Since both methods show a clear trade-off between the certified bound and corresponding robustness accuracy, we include the certified bounds and corresponding accuracy in parenthesis for both models, along with the accuracy on natural examples.

Table 1: Comparison on MNIST and CIFAR-10. The numbers "$a\ (b\%)$" mean a certified bound $a$ with the corresponding accuracy $b\%$.

|  | MNIST | | CIFAR-10 | |
|---|---|---|---|---|
| Model | Robust Accuracy | Natural | Robust Accuracy | Natural |
| [17] (Single) | 1.58 (43.5%) | 88.2% | 36.00 (53.0%) | 61.2% |
| [17] (Cascade) | **1.58 (74.6%)** | 81.4% | 36.00 (58.7%) | 68.8% |
| STN | 1.58 (69.0%) | **98.9%** | **36.00 (65.6%)** | **80.5%** |

Our bound is close to the one from [17] on MNIST, and becomes better on CIFAR-10. In addition, since the training objective of [17] is particularly designed for provable robustness and depends on the size of attacks, its accuracy on the natural examples decreases drastically when accommodating stronger attacks. On the other hand, STN is capable of keeping a high accuracy on natural examples while providing strong robustness guarantees.

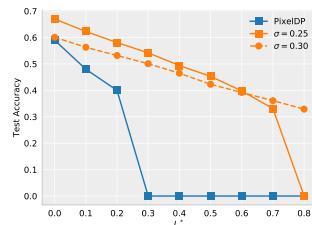

Figure 2: Comparison of the certified bound from STN (orange) and PixelDP (blue) on ImageNet.

**Certified Robustness on ImageNet** As our framework adds almost no extra computational burden on the training procedure, we are able to compute accuracy lower bounds for ImageNet [37], a large-scale image dataset that contains over 1 million images and 1,000 classes. We compare our bound with PixelDP in Figure 2. Clearly, our bound is higher than the one obtained via PixelDP.

## 6.2 Empirical Results

We next perform classification and measure the accuracy on real adversarial examples to evaluate the empirical performance of our defense methods. For each pair of attacks and defense models, we generate a robust accuracy vs. perturbation size curve for a comprehensive evaluation. We compare our method to TRADES on MNIST and CIFAR-10. Although we have emphasized the theoretical bound of the defense, the empirical performance is promising as well. More details and results of the experiments, such as for gradient-free attacks, are included in the Appendix.

**Avoid Gradient Masking** A defensive model incorporating randomness may make it difficult to apply standard attacks, by causing gradient masking as discussed in [10], thus achieving robustness unfairly. To ensure the robustness of our approach is not due to gradient masking, we use the expectation of the gradient with respect to the randomization when estimating gradients, to ensemble over randomization and eliminate the effect of randomness, as recommended in [10, 38]. In particular, the gradient is estimated as $\mathbb{E}_{\mathbf{r} \sim N(0, \sigma^2 I)}\left[\nabla_{\mathbf{x}+\mathbf{r}} L(\theta, \mathbf{x} + \mathbf{r}, y)\right] \approx \frac{1}{n_0} \sum_{i=1}^{n_0}\left[\nabla_{\mathbf{x}+\mathbf{r}_i} L(\theta, \mathbf{x} + \mathbf{r}_i, y)\right]$, where $\mathbf{r}_i$'s are i.i.d. samples from $N(\mathbf{0}, \sigma^2 I)$ distribution, and $n_0$ is the number of samples. We assume threat models are aware of the value of $\sigma$ in Algorithm 1 and use the same value for attacks.

**White-box Attacks** For $\ell_\infty$ attacks, we use Projected Gradient Descent (PGD) attacks [26]. It constructs an adversarial example by iteratively updating the natural input along with the sign of its gradient and projecting it into the constrained space, to ensure its a valid input. For $\ell_2$ attacks, we perform a Carlini & Wagner attack [8], which constructs an adversarial example via solving an optimization problem for minimizing distortion distance and maximizing classification error. We also use technique from [39], that has been shown to be more effective against adversarially trained model, where the gradients are estimated as the average of gradients of multiple randomly perturbed samples. This brings a variant of Carlini & Wagner attack with the same form as the ensemble-over-randomization mentioned above, therefore it is even more fair to use it. The results for white-box attacks are illustrated in Figure 3.

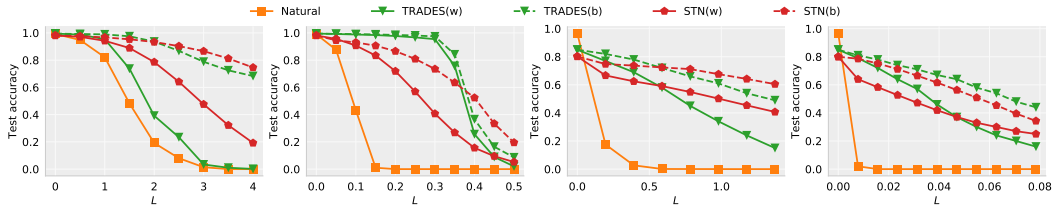

Figure 3: **MNIST and CIFAR-10**: Comparisons of the adversarial robustness of TRADES and STN with various attack sizes for both $\ell_2$ and $\ell_\infty$. The plots are ordered as: MNIST($\ell_2$), MNIST($\ell_\infty$), CIFAR-10($\ell_2$), and CIFAR-10($\ell_\infty$). Both white-box (straight lines) and black-box attacks (dash lines) are considered.

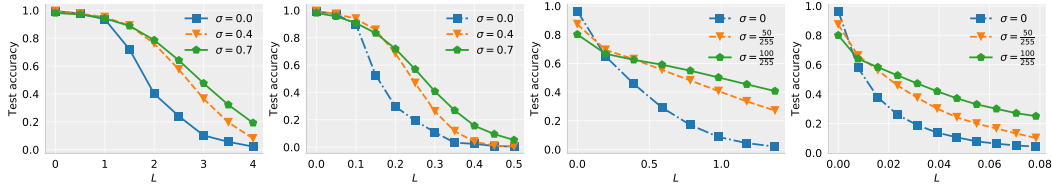

Figure 4: Robust accuracy of STN with different choices of $\sigma$ for both $\ell_2$ and $\ell_\infty$ attacks. The plots are ordered as: MNIST($\ell_2$), MNIST($\ell_\infty$), CIFAR-10($\ell_2$), and CIFAR-10($\ell_\infty$).

**Black-box Attacks** To better understand the behavior of our methods and to further ensure there is no gradient masking, we include results for black-box attacks. After comparison, we realize the adversarial examples generated from Madry's model [26] result in the strongest black-box attacks for both TRADES and STN. Therefore, we apply the $\ell_2$ and $\ell_\infty$ white-box attacks to Madry's model and test the resulting adversarial examples on TRADES and STN. The results are reported as the dashlines in Figure 3.

**Summary of Results** Overall, STN shows a promising level of robustness, especially regarding $\ell_2$-bounded distortions, as anticipated. One observes that STN performs slightly worse than TRADES when the size of attacks is small, and becomes better when the size increases. Intuitively, the added random noise dominantly reduces the accuracy for small attack size and becomes beneficial against stronger attacks. It is worth-noting that Algorithm 1 adds almost no computational burden, as it only requires multiple forward passes, and stability training only requires augmenting randomly perturbed examples. On the other hand, TRADES is extremely time-consuming, due to the iterative construction of adversarial examples.

**Choice of $\sigma$** In previous experiments, we use $\sigma = 0.7$ and $\sigma = \frac{100}{255}$ for MNIST and CIFAR-10, respectively. However, the choice of $\sigma$ plays an important role, as shown in Figure 1; therefore, we study in Figure 4 how different values of $\sigma$ affect the empirical robust accuracy. The results make it more clear that the noise hurts when small attacks are considered, but helps against large attacks. Ideally, using an adaptive amount of noise, that lets the amount of added noise grow with the size of attacks, could lead to better empirical results, yet it is practically impossible as the size of attacks is unknown beforehand. In addition, we include results for $\sigma = 0$, which is equivalent to a model without additive Gaussian noise. Its vulnerability indicates the essential of our framework is the additive Gaussian noise.

## 7 Comparison to [40]

Following this work, [40] proposed a tighter bound in $\ell_2$ norm than the one in section 4. Although they do indeed improve our bound, our work has unique contributions in several ways: $(i)$ We propose stability training to improve the bound and robustness, while they only use Gaussian augmentation. In general, stability training works better than Gaussian augmentation, as shown in Figure 1. Thus, stability training is an important and unique contribution of this paper. $(ii)$ We conduct empirical evaluation against real attacks and compare to the state-of-the-art defense method (adversarial training) to show our approach is competitive. [40] only discusses the certified bound and does not provide evaluation against real attacks. $(iii)$ The analysis from [40] is difficult to be extended to other norms, because it requires isotropy. On the other hand, ours lead to a tight certified $\ell_1$ bound by adding Laplacian noise, as discussed in Section 4.

## 8 Conclusions

We propose an analysis for constructing defensive models with certified robustness. Our analysis leads to a connection between robustness to adversarial attacks and robustness to additive random perturbations. We then propose a new strategy based on stability training for improving the robustness of the defense models. The experimental results show our defense model provides competitive provable robustness and empirical robustness compared to the state-of-the-art models. It especially yields strong robustness when strong attacks are considered.

There is a noticeable gap between the theoretical lower bounds and the empirical accuracy, indicating that the proposed upper bound might not be tight, or the empirical results should be worse for stronger attacks that have not been developed, as has happened to many defense models. We believe each explanation points to a direction for future research.

**Acknowledgments** This research was supported in part by DARPA, DOE, NIH, NSF and ONR.

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
