[Supplementary Material · adversarial_nips append.pdf]

# A Comparison between Bounds

We first use simulation to show our proposed bound is higher than the one from PixelDP [18].

In PixelDP, the upper bound for the size of attacks is indirectly defined: if $p_{(1)} \geq e^{2\epsilon} p_{(2)} + (1 + e^{\epsilon})$, where $\epsilon > 0$ and $\delta > 0$ are two tuning parameters, and the added noise has the distribution $N(0, \sigma^2 I)$, then the classifier is robust to attacks whose $\ell_2$ size is less than $\frac{\sigma\epsilon}{\sqrt{2\log(1.25/\delta)}}$.

As both our and their bound are determined by the models and data only through $p_{(1)}$ and $p_{(2)}$, it is sufficient to compare them with simulation for different $p_{(1)}$ and $p_{(2)}$ as long as $p_{(1)} \geq p_{(2)} \geq 0$, $p_{(1)} + p_{(2)} \leq 1$ and $p_{(1)} + p_{(2)} \geq 0.2$ are satisfied, $i.e.$, $p_{(1)}$ and $p_{(2)}$ are valid first and second largest output probabilities.

For fixed $\sigma$, $\epsilon$ and $\delta$ are tuning parameters that affect the result. For a fair comparison, we use a grid search to find $\epsilon$ and $\delta$ that maximizes their bound.

Figure 5: The upper bounds under different $p_{(1)}$ and $p_{(2)}$. Our bound (red) is strictly higher than the one from PixedDP (blue).

The simulation result in Figure 5 shows our bound is strictly higher than the one from PixelDP. In particular, when $p_{(1)}$ and $p_{(2)}$ are far apart, which is the most common case in practice, our bound is more than twice as high as theirs.

# B Proof of Lemma 1

**Lemma 1** Let $P = (p_1, \ldots, p_k)$ and $Q = (q_1, \ldots, q_k)$ be two multinomial distributions over the same index set $\{1, \ldots, k\}$. If the indexes of the largest probabilities do not match on $P$ and $Q$, that is $\text{argmax}_i \, p_i \neq \text{argmax}_j \, q_j$, then

$$D_\alpha(Q\|P) \geq -\log\left(1 - p_{(1)} - p_{(2)} + 2\left(\frac{1}{2}\left(p_{(1)}^{1-\alpha} + p_{(2)}^{1-\alpha}\right)\right)^{\frac{1}{1-\alpha}}\right) \tag{2}$$

where $p_{(1)}$ and $p_{(2)}$ are the largest and the second largest probabilities in $p_i$'s.

**Proof** Think of this problem as finding $Q$ that minimizes $D_\alpha(Q\|P)$ such that $\text{argmax} p_i \neq \text{argmax} q_i$ for fixed $P = (p_1, \ldots, p_k)$. Without loss of generality, assume $p_1 \geq p_2 \geq \cdots \geq p_k$.

It is equivalent to solving the following problem:

$$\min_{\sum q_i = 1, \text{argmax} q_i \neq 1} \frac{1}{1-\alpha} \log\left(\sum_{i=1}^{k} p_i \left(\frac{q_i}{p_i}\right)^\alpha\right)$$

As the logarithm is a monotonically increasing function, we only focus on the quantity $s(Q\|P) = \sum_{i=1}^{k} p_i \left(\frac{q_i}{p_i}\right)^{\alpha}$ part for fixed $\alpha$.

We first show for the $Q$ that minimizes $s(Q\|P)$, it must have $q_1 = q_2 \geq q_3 \geq \cdots \geq q_k$. Note here we allow a tie, because we can always let $q_1 = q_1 - \epsilon$ and $q_2 = q_2 + \epsilon$ for some small $\epsilon$ to satisfy $\text{argmax} q_i \neq 1$ while not changing the Renyi-divergence too much by the continuity of $s$.

If $q_j > q_i$ for some $j \geq i$, we can define $Q'$ by mutating $q_i$ and $q_j$, that is $Q' = (q_1, \ldots, q_{i-1}, q_j, q_{i+1} \ldots, q_{j-1}, q_i, q_{j+1}, \ldots, q_k)$, then

$$
s(Q\|P) - s(Q'\|P)
$$
$$
= p_i \left(\frac{q_i^{\alpha} - q_j^{\alpha}}{p_i^{\alpha}}\right) + p_j \left(\frac{q_j^{\alpha} - q_i^{\alpha}}{p_j^{\alpha}}\right)
$$
$$
= (p_i^{1-\alpha} - p_j^{1-\alpha})(q_i^{\alpha} - q_j^{\alpha}) > 0
$$

which conflicts with the assumption that $Q$ minimizes $s(Q\|P)$. Thus $q_i \geq q_j$ for $j \geq i$. Since $q_1$ cannot be the largest, we have $q_1 = q_2 \geq q_3 \geq \cdots \geq q_k$.

Then we are able to assume $Q = (q_0, q_0, q_3, \ldots, q_k)$, and the problem can be formulated as

$$
\min_{q_0, q_2, \ldots, q_k} p_1 \left(\frac{q_0}{p_1}\right)^{\alpha} + p_2 \left(\frac{q_0}{p_2}\right)^{\alpha} + \sum_{i=3}^{k} p_i \left(\frac{q_i}{p_i}\right)^{\alpha}
$$
$$
\text{subject to} \quad 2q_0 + q_3 + \cdots + q_k = 1
$$
$$
\text{subject to} \quad q_i - q_0 \leq 0 \quad i \geq 1
$$
$$
\text{subject to} \quad -q_i \leq 0 \quad i \geq 0
$$

which forms a set of KKT conditions. Using Lagrange multipliers, one can obtain the solution $q_0 = \frac{q^*}{1-p_1-p_2-2q^*}$ and $q_i = \frac{p_i}{1-p_1-p_2-2q^*}$ for $i \geq 3$, where $q^* = \left(\frac{p_1^{1-\alpha} + p_2^{1-\alpha}}{2}\right)^{\frac{1}{1-\alpha}}$.

Plug in these quantities, the minimized Renyi-divergence is

$$
-\log \left(1 - p_1 - p_2 + 2 \left(\frac{1}{2}\left(p_1^{1-\alpha} + p_2^{1-\alpha}\right)\right)^{\frac{1}{1-\alpha}}\right)
$$

Thus, we obtain the lower bound of $D_{\alpha}(Q\|P)$ for $\text{argmax} p_i \neq \text{argmax} q_i$. ∎

## C  Proof of Theorem 2

A simple result from information theory:

**Lemma 4** *Given two real-valued vectors $\mathbf{x}_1$ and $\mathbf{x}_2$, the Rényi divergence of $N(\mathbf{x}_1, \sigma^2 I)$ and $N(\mathbf{x}_2, \sigma^2 I)$ is*

$$
D_{\alpha}(N(\mathbf{x}_1, \sigma^2 I)\|N(\mathbf{x}_2, \sigma^2 I)) = \frac{\alpha \|\mathbf{x}_1 - \mathbf{x}_2\|_2^2}{2\sigma^2} \tag{3}
$$

**Theorem 2** Suppose we have $\mathbf{x} \in \mathcal{X}$, and a potential adversarial example $\mathbf{x}' \in \mathcal{X}$ such that $\|\mathbf{x} - \mathbf{x}'\|_2 \leq L$. Given a k-classifier $f : \mathcal{X} \to \{1, \ldots, k\}$, let $f(\mathbf{x} + N(\mathbf{0}, \sigma^2 I)) \sim (p_1, \ldots, p_k)$ and $f(\mathbf{x}' + N(\mathbf{0}, \sigma^2 I)) \sim (p_1', \ldots, p_k')$.

If the following condition is satisfied, with $p_{(1)}$ and $p_{(2)}$ being the first and second largest probabilities in $p_i$'s:

$$
\sup_{\alpha > 1} \left(-\frac{2\sigma^2}{\alpha} \log \left(1 - 2M_1\left(p_{(1)}, p_{(2)}\right) + 2M_{1-\alpha}\left(p_{(1)}, p_{(2)}\right)\right)\right) \geq L^2 \tag{4}
$$

then $\text{argmax}_i p_i = \text{argmax}_j p_j'$

**Proof** From lemma 4, we know for $\mathbf{x}$ and $\mathbf{x}'$ such that $\|\mathbf{x} - \mathbf{x}'\|_2 \leq L$, with a k-class classification function $f : \mathcal{X} \to \{1, \ldots, k\}$:

$$D_\alpha(f(\mathbf{x}' + N(\mathbf{0}, \sigma^2))\|f(\mathbf{x} + N(\mathbf{0}, \sigma^2)))$$
$$\leq D_\alpha(\mathbf{x}' + N(\mathbf{0}, \sigma^2)\|\mathbf{x} + N(\mathbf{0}, \sigma^2))$$
$$\leq \frac{\alpha L^2}{2\sigma^2}$$

if $N(\mathbf{0}, \sigma^2)$ is a standard Gaussian noise. The first inequality comes from the fact that $D_\alpha(Q\|P) \geq D_\alpha(g(Q)\|g(P))$ for any function $g$.

Therefore, if we have

$$- \log \left( 1 - 2M_1 \left( p_{(1)}, p_{(2)} \right) + 2M_{1-\alpha} \left( p_{(1)}, p_{(2)} \right) \right) \geq \frac{\alpha L^2}{2\sigma^2} \tag{5}$$

It implies

$$D_\alpha(f(\mathbf{x}' + N(\mathbf{0}, \sigma^2))\|f(\mathbf{x} + N(\mathbf{0}, \sigma^2)))$$
$$\leq - \log \left( 1 - 2M_1 \left( p_{(1)}, p_{(2)} \right) + 2M_{1-\alpha} \left( p_{(1)}, p_{(2)} \right) \right) \tag{6}$$

Then from Lemma 1 we know that the index of the maximums of $f(\mathbf{x} + N(\mathbf{0}, \sigma^2))$ and $f(\mathbf{x}' + N(\mathbf{0}, \sigma^2))$ must be the same, which means they have the same prediction, thus implies robustness. ∎

## D   Details and Additional Results of the Experiments

In this section, we explain the details of our implementation of our models and include additional experimental results.

### D.1   Gradient-Free methods

We include results for Boundary Attack [9] which is a gradient-free attack method. Boundary attack explores adversarial examples along the decision boundary using a rejection sampling approach. Their construction of adversarial examples do not require information about the gradient of models, thus is an important complement to gradient-based methods.

We test Boundary attacks on MNIST and CIFAR10 and compare them to other attacks considered.

Figure 6: **MNIST**: Comparisons the adversarial robustness of STN against various types of attacks for both $\ell_2$ (left) and $\ell_\infty$ (right).

From the plots, one can see Boundary attack is not effective in attacking our models. This is consistent with the observation from [38] that gradient-free method is not effective against randomized models. Nevertheless, we include the results as a sanity check.

Figure 7: **CIFAR-10**: Comparisons the adversarial robustness of STN against various types of attacks for both $\ell_2$ (left) and $\ell_\infty$ (right).