[Reviews · NeurIPS 2019]

Reviewer 1



*** FURTHER UPDATE: After discussion with the AC, I am happy to change my score to 5 as we agreed the novelty of this paper should be judged before the publication made by Cohen et al. *** UPDATE: I thank the authors for their response. I have just some further comments. The certified bound for L_infty=0.3 for MNIST shown in Figure 2 shows that it is approximately 70% accuracy? Whereas TRADES seems to be closer to 100% and Gowal et al is above 90% - it seems low compared to the numbers I am used to. This might be due to the bound being too loose. I definitely agree that the goal of the adversary is to find an image where the difference is imperceptible to the human eye, however, when the perturbation radius is larger we should be less sure that **all** images within this space are imperceptible to the original. To test gradient obfuscation just looking at black box attacks is often not enough. Here I would like to refer the authors to the paper: Obfuscated Gradients Give a False Sense of Security: Circumventing Defenses to Adversarial Examples - Section 3.1. Again I thank the authors for the response, but given the concerns I still have I will keep my original score. *** Overall I found the paper slightly hard to read, my main comments are below: - The initial figure (Figure 1) has a mismatching caption and seems to be missing some curves that are referenced below ? It is unclear to me how to interpret the figure. The caption mentioned that there should be a pink line which shows TRADES, but I cannot find a pink line on the plot. The brown line and purple line I guess should be the green and orange line shown shown in the Figure? The difference between the caption and the figure itself makes the results presented very confusing. On top of this the experimental section in general is hard to follow, weakening the paper. For example in Table 1 the authors list a result of “1.58 (69.0%)” this is either suggesting a certified bound of 1.58 in an unclear metric (with corresponding accuracy of 69%) which would not be a very good result, or it suggests a robustness bound (measured in error) in which case the accuracy contradicts the bound. - The results are not good compared to state of the art and it is unclear how the theoretical result can help to push the boundaries of current research going on in the field. For example: ‘To interpret, the results, for example on MNIST, when sigma=0.7, Algorithm 1 achieves at least 51% accuracy under any attack whose l2_norm size is smaller than 1.4.’ This seems to be well off the state of the art as Gowal et al: ‘On the effectiveness of Interval Bound Propagation for Training Verifiably Robust Models’ which has verified accuracy on MNIST 92% on l_infinity norm of 0.3. - Figure 2 shows that TRADES gives better results for L infinity norm considered, more worryingly - STN does not seem to perform well when the epsilon is small (namely when the perturbation is most imperceptible) for L2 norm. The authors also made a comment on the fact that TRADES is expensive to train - TRADES used 10 FGSM steps to train the network, this means that it would be approximately 10X more expensive than standard ERM. The authors alluded to the fact that their method is cheaper, can the authors quantify how much cheaper it is in comparison to TRADES? -The authors also talked about gradient masking - here the comments were made about what they did to combat gradient masking but there is no strong empirical evidence that the gradient masking is indeed not occurring. In order to do this, can the authors show the adversarial accuracy of the networks trained under a very strong attack - e.g. PGD 1000 steps and maybe 20 different random initializations of the initial perturbation. Other comments: - Can the authors comment if this theoretical result can be extended to other L_p norms? The authors alluded to the fact the theoretical lower bound is not tight in the conclusion, how would this change if we consider other norms? - A side note, I would prefer the figures to have corresponding Tables in the supplementary, this allows us to see the exact adversarial accuracy achieved. - A general note in talking about adversarial accuracy under PGD. The authors should always report the step size and the number of PGD steps performed as the adversarial accuracy is highly dependent upon these parameters. In fact a lot more details are needed to describe the attack: if you are using PGD, is the optimizer you are using just standard SGD or Momentum or Adam? To know how strong your results are you will need to specify these parameters so others can have an idea of how strong the attack is.

Reviewer 2



The paper studies a novel method for making neural networks robust to norm bounded adversarial perturbations, in particular obtaining provable guarantees against perturbations in the L2 norm and empirical results showing that the networks are also robust to perturbations in the Linf norm. The paper is well written and the algorithmic and theoretical contributions are clearly outlined. My specific concerns center around the originality of the paper (distinction from prior work) and quality of some of the experimental results obtained. Originality: The paper improves upon the analysis from Lecuyer et al and develops a training method for producing certifiably robust randomized classifiers that is novel as far as I know. However, it is unclear to me how much the paper improves upon the analysis of Lecuyer and what the main source of the improvement was. Further, there is more recent work by Cohen et al that appeared on ArXiv https://arxiv.org/abs/1902.02918 in February (and a revised version was later published at ICML). I understand that the authors may have missed this work at the time of submission, but since the paper was already online before the submission deadline and published soon after, I would appreciate clarifications from the authors regarding the novelty in the rebuttal phase. In particular, I would like to understand: 1) The developed certificates only apply to L2 norm perturbations. Can the authors' framework providing any guarantees for perturbations in other norms? 2) How does the certificate derived compare to that in Lecuyer et al? In particular, is the certificate always guaranteed to be tighter than the one from Lecuyer et al? If not, what are the regimes where it is tighter? 3) How does the work compare to that of Cohen et al? Cohen et al claim that their certificate is the tightest one can obtain for the L2 norm for binary classifiers given that the only information known about the classifier is the probability of correct classification under random Gaussian perturbations. Given this, how does the certificate derived by the authors compare? Quality: The proofs of the mathematical results are correct in my assessment. The experimental results are interesting and indicate that the method developed indeed produces provably robust classifiers. However, there are a few issues with the experimental evaluation I wanted to clarify: 1) The authors mention that they multiply the confidence interval by 95% to obtain the corresponding accuracy. This seems very confusing to me. What accuracy is referred to here? If this is the accuracy in terms of the fraction of test examples certified to be adversarially robust, I find this a bit confusing, since the two probability spaces (sampling over the data distribution vs sampling over the Gaussian perturbations) are unconnected. 2) The improvements over PixelDP seem to come largely from the training method. (ie the blue curves are much above the orange curves in figure 1). What if the PixelDP bound was evaluated on the classifier trained by STN? 3) Table 1 is rather confusing. I assume the certified bound is the radius of the perturbation being certified. Why was this specific value chosen? Was it to maintain consistency with [17]? 4) Since obtaining certificates on ImageNet is a significant advance, I would advise the authors to include these results in the main paper rather than the appendix. Clarity: The paper is well written overall and the details are clear and easy to follow. Significance: I think

Reviewer 3



***EDITED REVIEW*** I thank the authors for addressing my concerns in the rebuttal. I thank the authors for including a comparison to Cohen et al and I am surprised to see that their method outperforms Cohen et al. Cohen et al argue that it is not possible to surpass their bound with a randomized smoothing approach. If indeed stability training is that much more powerful than Gaussian data augmentation, I think the results are interesting to the adversarial community, since Gaussian data augmentation is still a very common baseline (e.g. https://arxiv.org/abs/1906.08988, https://arxiv.org/abs/1901.10513 and others). The authors acknowledge that Cohen et al improve their bound; therefore, the adversarial community would benefit most from a combination of Cohen's better bound and stability training. For this reason, the paper should be rewritten, since the main contribution stems from stability training. However, stability training itself is not a novel contribution as it was introduced in "Improving the robustness of deep neural networks via stability training" and it was already demonstrated in reference [32] of this paper that stability training works better than Gaussian noise for increasing performance on noise corruptions. Due to the Discussion with another reviewer, I agree that the results on CIFAR-10, l_inf are not convincing and there are other methods that are faster than TRADES that achieve better results (e.g. https://arxiv.org/pdf/1811.09716). I have changed my score to a 6, because I find it interesting that the results of Cohen et al are outperformed. I do think that the authors should rewrite the paper focussing more on the improvement due to stability training and maybe use Cohen's better bound. ******** The authors derive and prove adversarial robustness bounds based on the Renyi divergence and utilizing stability training. They build upon the methods presented in Lecuyer et al [2018] and show higher robustness bounds. The paper is written in the context of randomization-based defenses where noise is added to inputs; if the inputs have been perturbed adversarially before that, the predictions of a classifier are smoothened by the added noise. Originality, Quality and Significance: My main comment is that the current state of the art in certified adversarial robustness, namely Cohen et al. [2019] was not discussed and not compared to. Cohen et al. use a very similar approach as they also add Gaussian noise to adversarial examples to “drown out” adversarial perturbations and even prove that it is impossible to certify higher l2-robustness with a radius larger than their derived R. In more detail, Cohen et al use Gaussian data augmentation during training, while this paper uses stability training where an additional stability term is included during training. The stability term is the cross-entropy loss between f(x) and f(x’) where x is clean data and x’ is data with added Gaussian noise. The original publication on stability training [Zheng et al, 2016] reported that data augmentation with Gaussian noise leads to underfitting compared to their stability training approach. Another recent study demonstrates better model performance of stability training compared to data augmentation and also decreased sensitivity to hyperparameters [Laermann et al, 2019]. I would be interested to see whether this difference (stability training is better than data augmentation) leads to a better robustness bound compared to the approach using data augmentation. It would be crucial for the authors of this paper to compare to Cohen et al. as the approaches are very similar and it is not clear whether the approach of Cohen et al. can be superseded at all with another randomization-based method due to the tight bounds argument of Cohen et al. Additionally, Cohen et al. provide results for ImageNet and the authors should include them in their discussion/ experiments’ section. Instead, the authors compare their results to Wong et al. [2018] which is a baseline that was already beaten by Cohen et al. If the results from comparing to Cohen et al are worse, I would suggest rejecting this paper as the approaches are very similar. On the other hand, the authors achieve better performance than TRADES on l2, both for MNIST and CIFAR10 with a less time-consuming approach. Cohen et al have not compared their method with TRADES, so it is not clear which would perform better. In case this approach performs better than Cohen et al (and as already shown better than TRADES), the paper would be worth accepting. Clarity: The paper is well written and it is easy to read. The form of the presentation is therefore good and acceptable for NeurIPS. I have several minor comments. In Figure 1, the colors in the caption do not match the colors in the image. In Table 1, I could not find the cited results in Wong et al. [2018]. In Wong et al. [2018], in Table 4, the results look similar to the numbers in Table 1, but they are not exact. For example, in Table 1, Cifar-10, Single, the robust accuracy is displayed as 53% which would correspond to an error of 47%. This robust accuracy however does not occur in Table 4, Wong et al. [2018]. The natural accuracy seems off by 10% for CIFAR-10. The other numbers are also slightly off. Did the authors of this paper use the code of Wong et al. [2018] to rerun their simulations which naturally results in slightly different numbers? Also: Which model was used in this comparison as there are 3 different models both for CIFAR-10 and for MNIST in Wong et al. [2018], in Table 4. In Figure 2, the orange curve displaying the Natural case. Please state whether this is a vanilla trained model and how it was attacked. There are some typos in the text, but not many, such as eg. Page 3, line 110: ‘robustss’ instead of robustness. In the Appendix, there are a few typos in D1 such as in the sentence: ‘Since only PixelDP [18] is able to obtain non-trivial certified bound on ImageNet, we compare out bound to theirs’ -> 1) obtain a non-trivial… and 2) our bound. References: Lecuyer, Mathias, et al. "On the Connection between Differential Privacy and Adversarial Robustness in Machine Learning." arXiv:1802.03471 (2018). Cohen, Jeremy, et al. “Certified Adversarial Robustness via Randomized Smoothing”, arXiv:1902.02918 (2019). Wong, Eric et al. „ Scaling provable adversarial defenses”, arXiv:1805.12514 [2018] Zheng, Stephan et al., “Improving the Robustness of Deep Neural Networks via Stability Training”, https://arxiv.org/pdf/1604.04326 [2016] Laermann, Jan et al. “Achieving Generalizable Robustness of Deep Neural Networks by Stability Training” https://arxiv.org/pdf/1906.00735 [2019]

[Author Response · NeurIPS 2019]

We appreciate the valuable comments from all reviewers. We first respond to common issues brought by the reviewers, and then respond to individual comments.

**Comparison to Cohen et al, [2019]:** Both Reviewers 2 and 3 asked for a comparison to Cohen et al. We did not include such a comparison because their work is actually a follow-up of our work. As a work released earlier (posted on arXiv), we thought it was not necessary to include all follow-up works in comparisons. Although they do indeed improve our bound by a small margin, our work has novel contributions in several ways: 1) We propose stability training to improve the bound and robustness, while they only use Gaussian augmentation. As pointed out by Reviewer 3, stability training works better than Gaussian augmentation. The improvement from stability training is more significant than the improvement of the bound. As a result, our bound plus stability training could yield a higher certified bound and empirical robustness accuracy than Cohen et al. We show the improvement of the bound and robust accuracy on CIFAR10 in Figure 1. The

Figure 1: Bounds & empirical robust accuracy comparisons on CIFAR10 with ours and Cohen et al.

gap in robust accuracy is more obvious as their tighter bound does not help improve robustness against real attacks. Thus, stability training is an important and unique contribution to the literature. 2) We conduct empirical evaluation against real attacks and compare to the state-of-the-art defense method to show our approach is competitive. Cohen et al. only discusses the certified bound and does not provide evaluation against real attacks. 3) The analysis from Cohen et al. is difficult to be extended to other norms, while ours can be easily achieved. Both Reviewer 1 and 2 asked about extension to other norms and we now explain. For example, noticing the Renyi divergence between two Laplacian distribution $\Lambda(x, \lambda)$ and $\Lambda(x', \lambda)$ is $\frac{\|x-x'\|_1}{\alpha-1} \log \left( \frac{\alpha}{2\alpha-1} e^{\frac{\alpha-1}{\lambda}} + \frac{\alpha-1}{2\alpha-1} e^{-\frac{\alpha}{\lambda}} \right)$, one can obtain a certified $\ell_1$ bound by adding Laplacian noise. The derivation is similar to the proof of Theorem 1 by replacing Gaussian with Laplacian. In general, our framework extends to any norm as long as we find a corresponding distribution whose Renyi divergence of two is a function of the distance of their means. Thus, our analysis provides better flexibility. We will discuss the differences more thoroughly in the revision.

**Caption of Figure 1:** We apologize for the confusion in the caption. We will revise it so the colours match the plots.

**Reviewer 1:** 1) The fundamental difference from TRADES is our approach provides certified robustness, *i.e.*, our model is theoretically guaranteed to be robust as long as the norm of the perturbation is smaller than the bound. Although TRADES is empirically robust, no theoretical guarantee is provided. The fact that our approach is empirically competitive to TRADES should be considered as a great bonus. 2) The certified bound from Gowal et al. is for $\ell_\infty$ norm which is not comparable to $\ell_2$ norm. The state-of-the-art $\ell_2$ certified robustness is achieved at Wong et al. [2018] in Table 4, to which we compare in our paper. 3) Our approach is much cheaper than TRADES. Training TRADES on Wide-Resnet for CIFAR10 takes more than 5 days on a T4 GPU; whereas our approach does not require constructing adversarial examples and only takes 8 hours to train in the same setting. Thus our method is about 15 times cheaper than TRADES. In fact, due to the computational constraint, TRADES is not scalable to ImageNet, while our approach is, as shown in Appendix D. 4) We argue the comment that we care more about the smaller perturbations than the large ones is not precise. The goal of adversaries is to reduce the classification as much as the perturbation is not perceptible. For example, if any perturbation smaller than 2.0 is not perceptible, the adversaries should always use 2.0 instead of any number smaller, because larger perturbations strictly reduce more accuracy. In this sense, larger perturbations are more important than smaller ones. 5) We use two commonly used gradient-free attacks (transfer attack and boundary attack) to evaluate the robustness of our method. We believe it is sufficient to show there is no gradient masking. We will add the results for PGD-1000 with 20 random initializations in the revised version. 6) In our evaluation, we use 20 steps PGD with step size $\alpha = \epsilon/10$. These parameters can be found in our code. We will make it more clear in the revised version.

**Reviewer 2:** 1) Our bound is strictly tighter than the one from Lecuyer et al. which is shown in Appendix A with simulation. We also evaluate PixelDP and our bound without STN on MNIST and CIFAR10 to show the empirical gap in Figure 1 in our paper (orange and green). 2) It is exactly because the two probabilities are not correlated that we can multiply them together. The probability that two independent events (bound is correctly estimated and the example is correctly classified) happen simultaneously is the product of their individual probabilities. 3) Yes, the radius is chosen to maintain consistency with Wong et al. 5) We will include results for ImageNet in the main text in the revised version.

**Reviwer 3:** 1) In Table 1, we are comparing to the best results (bold numbers) reported in Wong et al [2018]. For the single model on CIFAR10, the robust accuracy should be 52% instead 53% as they reported the robust error being 48%. Similarly, the natural accuracy for Cascade model on CIFAR10 should be 58.8% instead of 68.8%. We apologize for the mistakes, but in both cases the actual numbers are worse than what is reported. 2) The orange curve corresponds to a vanilla trained model attacked by PGD. We will make it more clear and fix the typos in the revised version.

[Meta-Review · NeurIPS 2019]

The paper proposes a certified defense method by adding Gaussian noise and a stability training approach to further improve robustness. After the rebuttal, the authors addressed most of the concerns and the reviewers agreed that this paper has some interesting contributions. Reviewer 1 still has some concerns about the comparison with TRADES and we hope the authors can address this in the final version. Also, although we are judging the novelty of this paper assuming there is no (Cohen et al), the authors should cite and discuss about (Cohen et al) in the camera ready version.